# Do Long-Run Disasters Promote Human Capital in China? —The Impact of 500 Years of Natural Disasters on County-Level Human-Capital Accumulation

**DOI:** 10.3390/ijerph17207422

**Published:** 2020-10-12

**Authors:** Zhidi Zhang, Jianqing Ruan

**Affiliations:** China Academy for Rural Development, Zhejiang University, Hangzhou 310058, China; 11722016@zju.edu.cn

**Keywords:** long-run disasters, human capital, quantified history, drought, flood

## Abstract

Is there a relationship between the frequency of regional natural disasters and long-term human-capital accumulation? This article investigates the long-run causality between natural calamities and human-capital accumulation with macro and micro data. Empirical cross-county analysis demonstrates that higher frequencies of natural calamities are correlated with higher rates of human-capital accumulation. Specifically, on the basis of empirical data of the fifth census in 2000 and China’s Labor-Force Dynamics Survey in 2012, this paper exploits the two databases to infer that the high disaster frequency in the years of 1500–2000 was likely to increase regional human-capital accumulation on district level. High natural-calamity frequency reduces the expected rate of returning to physical capital, which also serves to increase human-capital. Thus, experiencing with natural disasters would influence human’s preference to human-capital investment instead of physical capital.

## 1. Introduction

Beginning in the 2000s, economists began to consider more carefully the potential implications of long-run disasters for economic activity [1,2,3]. As for China, with 5000 years of farming culture, natural disasters have challenged human survival since ancient times, among which were droughts and floods, respectively, account for 57% and 30% of disasters [4]. Numerous studies on the relationship between natural disasters and expected destructions and losses are available and generally widely known [5,6], but to the best of our knowledge, there are no empirical studies that evaluate the effects of long-run disasters on human-capital accumulation in an agricultural economic framework.

There are, two distinct streams of literatures in political economy that examined the impacts of long-run natural disasters on human capital. One view is that infrastructure destroyed by natural disasters leads to a reduction in material capital investment in the short term, which increases the government’s debt burden. Thus, long-term sustained fiscal imbalances affect the development of human-capital investment [7]. Moreover, in countries with more frequent disasters, new foreign direct investment finds a wise investment environment [8]. At the same time, with a large amount of financial-capital transfer, affected areas bear the economic loss of capital flight, thereby reducing growth rate and causing labor loss. The opposite view is that natural disasters produce Schumpeterian creative destruction. Advanced technological equipment and technology are introduced during the reconstruction process and enhanced productivity then brings higher human-capital investment and productivity. Indeed, the process forms a virtuous circle that promotes an increase in human capital [9,10,11]. Aghion and Howitts [12] theoretically explained Schumpeterian creative destruction, which is a model with an endogenous economic-growth model of the innovative-destruction process. Meanwhile, Skidmore and Toya [9] used cross-country data research to place more emphasis on human-capital investment in places where natural disasters frequently occur. This process is equivalent to the increase in productivity and market competitiveness of production factors through technological innovation and capital substitution in the random events of disasters.

Our article extends the understandings of the linkages among long-run natural disaster and human capital to agricultural channels based on empirical evidences from China, Chinese people have suffered from severe famine, plague, poverty [5,6] and social-system turmoil [13,14], mainly due to reductions in food production caused by continuous droughts and floods. It is reasonable to question whether there exist some channels through agricultural factors. Based on the historical materials in China, we analyze the data relating to natural disasters in 1500–2000 from 120 prefectures (Figure 1) and proposed a positive relationship between regional natural disasters and local human-capital accumulation. Figure 1 reports the aggregated regional changes in the frequency of floods and droughts in China during period of 1500–2000, which overlaid with county boarders in 2000. With three levels disaster frequencies divided, huge floods and floods were concentrated in the central and southern parts of China, while Gansu, Sichuan, Guizhou and Yunnan provinces in the west had lower levels of floods and droughts.

Additionally, we demonstrate the psychological and cultural aspects of the channels that received few attentions in previous researches. When faced with risky disasters, people are more dichotomous in their behavior due to the pressure of emotions and fears [15]. In other words, the fear of disasters can affect people’s consumption expectations for fixed asset investment, which made human-capital accumulation awareness run continuously deep in the culture of this region [16]. As a general conventional society can only maintain the “Malthusian society” in the survival line, there were little assets that could be used to cushion when impacted by risk events. In this regard, self-protection behaviors such as “nurturing children to block the old” and “buying and selling wives” often occurred in ancient China which result in the increase of the relative value of human capital and the instability of capital and durable goods (Wuchang Zhang, 1972). Especially, due to the deeply rooted Chinese culture of “there are golden houses in books”, which encourages more human-capital investments whenever possible [17,18]. Parents can pass their human capital to their offspring through parenting [19]. To elaborate, Guo et al. [20] analyzed the role of traditional Chinese culture in human-capital accumulation, established a model of human-capital accumulation and refined the motivation of children’s education to examine the impact of intergenerational income transfer on human capital. Moreover, the academic emphasized that the motivation of parents for their children’s education is more preferred as “self-interest” motivation instead of “altruism” [21] and Chinese parents have attached great importance to their children’s academic success, so they are willing to invest much of money in their time and money for their children’s education [22,23]. In addition, small-probability and large-hazard incidents are not included in private insurance contracts. Although the government is trustworthy, more people can only bear disaster losses by themselves. Therefore, human capital increases continuously after a disaster due to risks and losses in material capital.

With natural calamities’ randomness and exogeneity, this paper adapts a series of linear regressions to test the effect of natural-calamity frequency on the local human capital. Funding that the county-level human-capital stock obtained from the 2000 census data is positively correlated with the frequency of county-level natural disasters in the Atlas of China’s Nearly Five Hundred Years of Drought and Flood Distribution, which is published by China Meteorological Administration Atlas of Drought and Flood Distribution in China. We present an initial evidence regarding to the relationship between the frequency of natural disasters and the per capita human-capital stock of each county in Figure 2. Results showed that individuals who experienced serious natural disasters were 1.594 units higher than those in the average family, while human capital at the household level was 1.522 units higher. Additionally, this article carries out empirical verification of both macro and micro data and provides theoretical explanations in the perspectives of psychology and local culture, which illustrates that human capital as continuous accumulation within the family, especially in the educated elite cohort.

The rest of this paper is organized as followed. In Section 2, this paper introduces the data and baseline models. Section 3 presents regression results and discusses the identification strategy used to evaluate the effect of the disaster frequencies on Jinshi regional admission rates, the 20 latest years and micro dataset form CLDS individual samples. Section 4 briefly concludes this paper.

## 2. Materials and Methods

### 2.1. Empirical Analysis

This section further empirically analyzes the extent to which the long-term frequency of natural calamities affects the accumulation of human capital. To study the impact of human-capital accumulation in the county, the empirical evidence of this section is divided into macro and micro parts. The data of disasters come from the published Atlas of Drought and Flood Distribution in China by China Meteorological Administration; human-capital data of the macro part are from the census in 2000, the macro part is from China’s Labor-Force Dynamic Survey in 2012 and control data are through regional economic analysis.

### 2.2. Frequency of Disasters

The frequency data of natural disasters in this paper come from the National Meteorological Administration’s Atlas of Drought and Flood Distribution in China and its extended data. The dataset contains drought and flood conditions of 120 meteorological observatories in China during the period from 1470 to 2000. This paper selected drought and flood conditions from 1500 to 2000 as independent variables to measure the frequency of natural disasters. The dataset uses a five-level system to indicate drought and flood conditions, where 1 indicates flooded, 2 indicates swamp, 3 indicates normal, 4 indicates dry and 5 indicates drought. The meteorological data points of 120 prefecture-level data were spatially interpolated with the ordinary Kriging method in ArcGIS Microsoft to obtain the average of county-level climate data. Since the calculation results contain decimals, this study assumed that the distribution of meteorological data in all counties across the country is consistent with 120 observatories that were restored into a five-level system. The interpolated data were spatially analyzed with county-level administrative regions of the country; the spatial distribution of disaster information in the county area was obtained and the average value was calculated. According to the above data [24,25], this paper calculated the frequency of droughts and floods, respectively. The weights of a severe and a general disaster were 2 and 1, respectively. The frequency of the historical disasters at the county level could be obtained by adding the average frequency of droughts and the floods. The severity of the 500 years varied across region. The Gansu, Sichuan, Guizhou and Yunnan provinces in the west have lower levels of floods and droughts, while more natural disasters are concentrated in the Shanxi, Shandong and Jiangsu provinces. Disparities in the effects of disaster frequency across regions are important components that are used to identify the long-term effects. The variation in disaster severity also had a different effect on human-capital stocks.

### 2.3. Human Capital in 2000

The measurement of human capital is difficult and complicated and there are many methods to do so. The data availability of the method of empirical education [26] is favored. This paper uses the method of Yue and Liu [27] to express the human-capital stock (HC) by using the product of the average age of education (h) and the amount of labor. Among them, the average age of education (h) of residents is based on the national population aged 6 years and over as the statistical caliber and the education level of residents is divided into five categories. This is a magnitude that may include college and above education, high school, junior high school, primary and illiteracy and semiliteracy, and the average cumulative years for each type of education were defined as 16, 12, 9, 6 and 0 years. The data came from the fifth census in 2000. Because the official sampling data of the years of education of workers in labor were only from 1996–1999, it was logical to use the education years of the entire population instead of the years of education of the practitioners. At the same time, the overall quality of a region’s population can capture the overall quality of the labor practitioners.

The formula for calculating the average number of years of education (hc) for residents is:(1)$$\mathrm{hc}=\frac{un_{school}\times 1+\mathrm{primary}\times 5+\mathrm{junior}\times 8+\mathrm{senior}\times 11+\mathrm{college}\times 15}{\mathrm{pop}}$$

In Formula (1), primary, junior, senior and college account for the proportion of the population aged 6 and over in primary, middle, high and college education, respectively.

However, the development of human capital is also pertinent to other factors, such as regional population and geographical area. Therefore, this study needs to further control other factors to empirically test the relationship between two core factors. On the basis of past research and the availability of data, this paper measured the human-capital stock of 2195 counties in the country and the frequency of natural disasters in 2195 counties. Table 1 is a descriptive statistic for the following main variables: (1) per capita GDP in the county area to examine the impact of the regional economy on human capital; (2) the proportion of minority population in the total population of the region; (3) the logarithm of the area (10,000 km^2^), which is the value of the area to investigate impact of the size when testing the damage to natural disasters; and (4) government fixed-asset investment in the county economy (CNY 10,000).

Summary statistics are presented in Table 1. The average district-level human-capital stock of China is 6.184 for 31 provinces and 0.584 of the disaster frequencies of the 2195 counties. Overall, the average district-level GDP in 2000 was CNY 0.714 billion and the average government fixed-asset investment was CNY 0.067 billion. In 2000, the average gender rate was 107.080 and the 12.53% of individuals in the census were ethnic minorities. The average logarithm of county areas was 7.623.

## 3. Results

The core part of this paper is to test whether natural disasters affect the development of county human-capital stock. Therefore, the dependent variable is the human-capital stock of each district and county. Thus, the key explanatory variable was the average frequency of flood and drought disasters during 1500–2000. As a climate disaster is an exogenous shock, especially in ancient days, it directly affects local agricultural outputs, for example, grain yield. Therefore, this paper is based directly on the following linear-regression specification:(2)$$\begin{array}{ll} HC_{i}=\beta_{0}+\beta_{1} & \cdot Disaster_{i}+\beta_{2}\cdot Ln{\left(pop\right)}_{i}+\beta_{3}\cdot Ethic\_rate_{i}+\beta_{4}\\ & \cdot Ln{\left(area\right)}_{i}+\beta_{5}\cdot PerInv\_cap_{i}+\beta_{6}\cdot North_{i}+{µ}_{i} \end{array}$$
where $HC_{i}$ meant the total human capital stock of the county *i* in 2000, $Disaster_{i}$ is the average of the frequency of natural disasters occurring in the county *i* area from 1500 to 2000. As discussed above, estimated coefficients, $\beta_{1}$ represent the effect of 500-year long-term disasters on reginal human-capital stock. This equation adds control variables that capture district-level socioeconomic factors, including county population, areas, ethnic rates and county-government fixed-asset investments. This formula also includes northern or southern region cohort fixed effects, $\beta_{6}$, defined by where the districts are located. Lastly, ${µ}_{i}$ is a random disturbance.

### 3.1. Long-Term Effects of Disaster Frequency—Empirical Results of Macro Data

Table 2 reports the results of the benchmark regression model. First, this paper used all the samples for analysis. The use of natural disasters as explanatory variables in Columns (1), (4) and (7) shows that the coefficient of natural disasters is significantly positive at the level of 1%; Columns (2), (5) and (8) control the area and population of the county, the proportion of ethnic minorities and the per capita investment in fixed assets. Results showed that, after the distance is factor controlled, the impact of natural disasters on human-capital stock is still significantly positive. In Columns (3), (6) and (9), this study further controlled the difference among political provinces; the coefficient of this variable was significantly positive, which indicated that natural disasters had an impact on capital stock except for the fixed effect among provinces. Ordinary-least-squares (OLS) regression results in Table 2 support the core view of this paper: long-term natural disasters can promote the evolution of human-capital stock.

Results show that, regardless of whether or not natural disasters are classified, their impact on human capital are significantly positive at the 1% confidence level. If natural disasters (1–3) are classified as floods (4–6) and droughts (7–9), their effects are greater than those of integrated types of natural disasters. In the OLS model (1), without control variables, the increase in the frequency of natural disasters has a positive impact on the human-capital stock of 2.160 units. If we add the control variables and control the provincial fixed effects (3), the frequency of natural disasters increases the human capital stock by 0.817 units. In terms of classification, natural types of droughts have greater impact on human capital than floods do. After adding the control variables and controlling the fixed effects of provinces (9), drought can affect 2.001 units, while the results of floods on local human-capital stocks are not significant. The effect of droughts on contemporary human capital was greater than that of floods.

### 3.2. Robustness

To ensure the robustness of the results, on the basis of the above empirical evidence, this returns the proportion of nominations of the nominees in the Ming and Qing Dynasties as indicators for measuring the human capital of the county. Regression removes disturbance factors such as urban time characteristics and conducts a robustness test. The relative fairness of the imperial examinations promoted the active participation of a wide range of people, enabled people to form an understanding of the importance of pursuing reading achievements and gave birth to a corresponding culture. Moreover, it has been intensifying and has been in existence for a long time. This culture of value-oriented education is reflected in the level of contemporary human capital. Ting [28] used the number of all scholars between 1368 and 1911 to measure the intensity of imperial examinations and used the average years of education in the 2010 census, the proportion of college students in the total population and the literacy rate to measure the level of human capital. This research found that there was very significant positive correlation between the two variables.

Migration could affect the interpretation of the results by changing the size and composition of the sample. If higher-educated individuals or households are more likely to migrate, then the result could be a sample with poorer human-capital stocks of in the region, as human-capital stocks are calculated by persons who attended schools. The choice of migration is always decided by the adults instead of teenagers attending schools. It should also be attempted to ascertain the extent of the of the district-level migration rate for 500 years. Unfortunately, as the long-term period lacks accurate data, this research does not have information on districts of migration.

As a robustness check, the Ming and Qing Dynasties Nominees Monument recorded a total of 51,624 people from the Ming and Qing Dynasties. The thesis could epitomize the characteristics and idiosyncrasy of the Ming and Qing dynasty elites’ human capital. These data are based on the First Collection of the Inscriptions on the Titles of the National Treasures published by Qing Emperor Qianlong in the eleventh year of the Qing Dynasty, as well as the rubbings of “Jinshi Titles and Monuments”, “Deng Ke Lu” and various local chronicles. Since the Qing Dynasty, trials were based on quotas set by prefectures and counties, the number of candidates could not be determined. In the 12th year of Shunzhi (1655), the number of people in the north and south was not revised and the number of selected people became flexible. However, due to the economic development and cultural-level gap between provinces, the number of scholars was not evenly distributed. Therefore, the proportion of the total number of nominees in counties and cities (1373–1904) had a representative role in the development of education in the region. Regression results showed that the frequency of natural disasters was significant for human capital everywhere. Results in Table 3 are consistent with results of previous partial empirical research. In (1) and (3) show that the coefficient of natural disasters and droughts are significantly positive at the level of 1%, respectively. With regard to (2), the coefficient of floods is not significant.

With the development and change of society, the impact of natural disasters in the past 20 years on contemporary people’s investment decisions is more direct. Therefore, the sample of 1980–2000 nearly 20 years, was added for this research. The results of the 20 years reflected in Table 4 were controlled after the provincial fixed effect and results are still significant, further illustrating the core view of human capital with long-term effects of natural disasters. When trim was applied to outliners of the main samples at 1st to 99th percentile (Table 5), R1 indicates that it was efficient and proper to predict the relationship among disasters and human-capital with 0.417 R^2^ value recorded. The result in R3 shows that trim droughts had a positive and significant effect on human-capital at 1% confidence level, which was the same with original linear-model result.

### 3.3. Test of Micro Survey Data

The measurement of human capital in the county area is the embodiment of indicators of the quality of the labor force. In this paper, through individual data of labor mobility, the CLDS of the Social Science Research Center of Sun Yat-sen University in 2012 verified the relationship between the frequency of natural disasters and human capital. The CLDS surveyed the entire labor force from 15 to 64 years old in the sample households. The individual labor and the survey communities were investigated on the basis of the random-stratified-sampling method. The measurement of human capital is the “type of formal education” in the individual questionnaire section. The degree of education is converted into years of education and the corresponding relationship is as follows: not over school = 0 years; primary school = 6 years, secondary school = 9 years; high school and specialist = 12 years; undergraduate = 16 years; master’s degree = 18 years; Ph.D. and above = 22 years. Other control variables mainly included several children in the family who were attending school, the total cost of schooling for their children and the educational level of their parents. Among them, the education level of parents was divided into 0–7 according to the type of formal education. Table 6 reports a statistical description of all variables in this section.

Summary statistics of analysis samples from the CLDS dataset are presented in Table 6. Overall, 4515 individuals had a 1.444 average gender rate with an average age of 39. During the period of 1500–2000, on an average of 0.597 disaster frequencies, the average education of samples was 8.734. The average samples’ father and mother education levels were 1.177 and 0.631, respectively.

The empirical model of this part refers to documents such as Goldstein [29] and Hofmann (1997) [30]. When there is a multilevel nested structure of sample data, the multilevel regression model should be used for analysis. For example, urban and rural areas are nested in provinces, communities (villages) are nested in urban and rural areas and families are nested in communities (villages). CLDS survey data selected in this paper have a hierarchical nesting structure because the survey subjects are divided into individuals, families and communities. Therefore, it is suitable to use a multilayer regression model (HLM). Therefore, the individual layer in this paper is used as the low-level unit of the model and the community layer is investigated as the high-level unit of the model. The model is as follows:(3)$$\begin{array}{c} hc_{j\xi }=\alpha_{0}+\alpha_{1}\cdot disaster_{j}+\alpha_{2}\cdot age_{j\xi }+\alpha_{3}\cdot N_{\mathrm{child}}{}_{j\xi }+\alpha_{4}\cdot father_{edu}{}_{j\xi }\\ \;+\alpha_{5}\cdot Mother_{j\xi }+\alpha_{6}\cdot educost_{j\xi }+\alpha_{7}\cdot edu\_num_{j\xi }+{µ}_{j}+r_{j\xi } \end{array}$$

Among them, $hc_{j\xi }$ refers to the human-capital stock represented by the highest degree of education obtained by individuals ξ in the survey; ${disaster}_{j}$ indicates the frequency of natural disasters occurring in the survey community; and ${µ}_{j}$ is the *j* survey community and overall intercept difference.

Regression results (Table 7) show that the main explanatory variables were significant and stable. From the perspective of microinvestigation, the level of education that has experienced serious natural disasters greatly increased. Results showed that individuals who had experienced serious natural disasters were 1.594 units higher than those in the average family, while human capital at the household level was 1.522 units higher. If classified according to the types of floods and droughts, the impact of droughts on human capital was relatively large at 3.710 units, while floods on human capital were only 2.211 units. On the household level, the results also had the same trend as that of the basic individual level. The impact of the droughts was 3.456 units, while the impact of floods was shown to be insignificant. Table 7 shows that the mixed regression results of the data hierarchy were not considered (4, 8, 12) and the impact of natural disasters on human capital is generally overestimated. The reason is that the same level of data comes from the same family, which contains the same family background, though, there is obvious heterogeneity between different individuals. At the same time, Table 8 reports the subsample results of classifying natural disasters into floods and droughts and also shows the robustness of the results of this study.

## 4. Conclusions

The analysis in this paper shows that long-term natural disasters provide an endogenous mechanism for intergenerational human-capital accumulation, thereby increasing the degree of innovation and human-capital stocks between regions. If the incidence of natural disasters in a region was high, there would be increasing difficulties needed to face for survival. In the context of highly devastating natural disasters on fixed capital investment, people continue to form the idea that investing in human capital is beneficial. Therefore, the formation of contemporary high human-capital regions is not an accident, but an inevitable result in historical evolution. This paper adds social and cultural factors to analyze problems in the field of economics and considers the evolution of culture from the perspective of long-term natural facts, which proves the origin of the increase in human capital between generations.

The main contributions of this paper are as follows: First, this paper proposed a new theory to explain the accumulation of human capital and carried out empirical verification of both macro and micro aspects. At the same time, it provides a theoretical explanation basis for natural disasters for the development of local culture. Third, there are endogenous problems in the study of human-capital stock and this paper used long-term historical disaster data as a dependent variable to explain the problem of avoiding endogeneity. Historical disasters cause large-scale migrations, which cause changes in human capital due to difficulties of obtaining tracking data. However, cross-section data used in this paper cannot reflect dynamic changes in the short term for disasters. These issues should be considered in the next study.

In conclusion, the incidence of natural disasters is directly related to the production income in rural areas, even though many factors affect human capital. Meanwhile, human capital is the endogenous driving force of economic development. The accumulation of human capital is not only affected by social systems and technological levels, but also the endogenous dynamics of long-term cultural development, which is directly related to the frequency of long-term natural disasters. Therefore, safeguarding the resilience of rural areas, promoting the accumulation of human capital in rural areas and paying attention to the health and education of farmers to improve the level of scientific and technological development are all potential drivers for China’s stable and balanced inter-regional development, which also could narrow the income gap between urban and rural areas.

## Figures and Tables

**Figure 1 ijerph-17-07422-f001:**
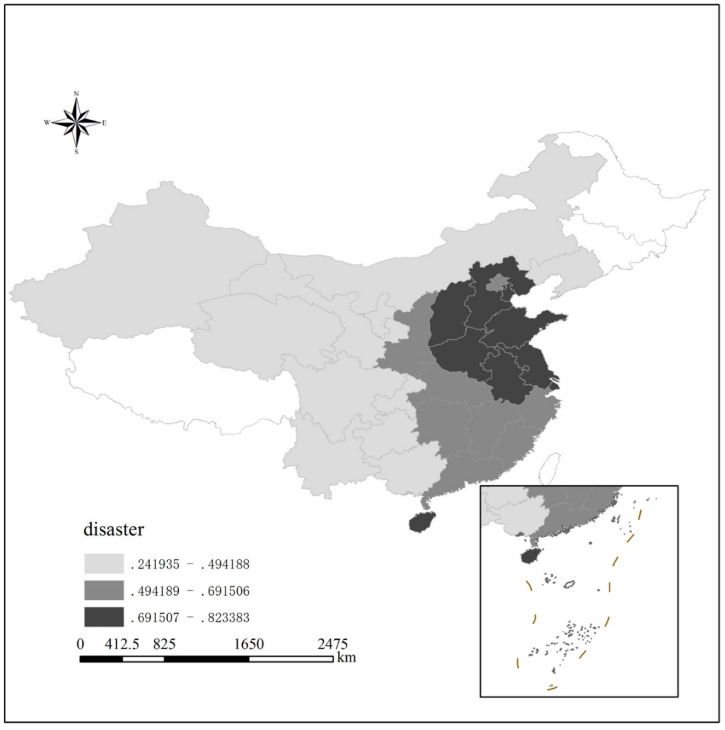
Regional differences in the frequency of flood and drought disasters in China during period of 1500–2000. Missing values shown as blanks in the figure.

**Figure 2 ijerph-17-07422-f002:**
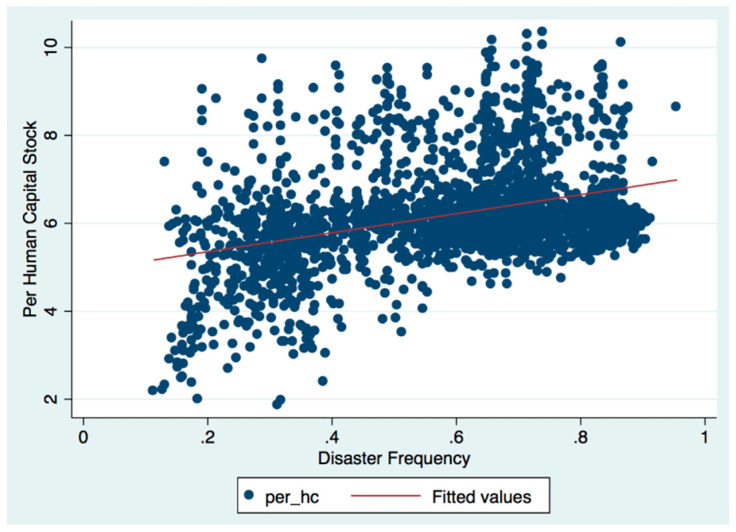
Scatter plot of the per capita human-capital stock of each county in China and the frequency of natural disasters.

**Table 1 ijerph-17-07422-t001:** Descriptive statistics for variables.

Variable	Definition	Sample	Mean	sd.	Min	Max
Per_hc	Human capital	2195	6.184	1.197	1.845	10.338
Disaster	Disaster frequency	2195	0.584	0.199	0.114	0.916
Per_gdp	Per GDP	2195	0.714	1.156	0.031	29.711
Gender	Female/male	2195	107.080	5.342	84.210	169.360
Ethic_rate	Ethnic population rate	2195	12.530	25.210	0	99.280
Ln(area)	Logarithm of the county area	2195	7.623	0.706	2.708	11.357
Per_capin	Per fixed investment	2195	0.067	0.118	0.001	2.503

**Table 2 ijerph-17-07422-t002:** Original linear model of frequency of disasters and human capital, ordinary-least-squares (OLS) estimates.

Variable	R1	R2	R3	R4	R5	R6	R7	R8	R9
Disaster	2.160 ***	1.250 ***	0.817 ***	-	-	-	-	-	-
	(0.131)	(0.143)	(0.260)	-	-	-	-	-	-
Flood	-	-	-	3.800 ***	2.074 ***	0.355	-	-	-
	-	-	-	(0.229)	(0.261)	(0.462)	-	-	-
Drought	-	-	-	-	-	-	3.251 ***	1.778 ***	2.001 ***
	-	-	-	-	-	-	(0.256)	(0.243)	(0.436)
Per_gdp	-	0.177 ***	0.148 ***	-	0.172 ***	0.150 ***	-	0.191 ***	0.144 ***
	-	(0.051)	(0.030)	-	(0.051)	(0.030)	-	(0.051)	(0.030)
Gender	-	−0.027 ***	−0.022 ***	-	−0.026 ***	−0.022 ***	-	−0.030 ***	−0.021 ***
	-	(0.005)	(0.004)	-	(0.005)	(0.004)	-	(0.005)	(0.004)
Ethic_rate	-	−0.017 ***	−0.018 ***	-	−0.016 ***	−0.019 ***	-	−0.019 ***	−0.019 ***
	-	(0.001)	(0.001)	-	(0.001)	(0.001)	-	(0.001)	(0.001)
Lnarea	-	0.302 ***	0.201 ***	-	0.286 ***	0.193 ***	-	0.287 ***	0.205 ***
	-	(0.034)	(0.032)	-	(0.033)	(0.032)	-	(0.034)	(0.032)
Per_capin	-	−0.088	−0.064	-	−0.027	−0.126	-	−0.279	−0.031
	-	(0.259)	(0.297)	-	(0.258)	(0.297)	-	(0.255)	(0.296)
FE—Province	NO	NO	YES	NO	NO	YES	NO	NO	YES
Constant	4.922 ***	6.096 ***	8.459 ***	5.043 ***	6.218 ***	9.052 ***	5.260 ***	6.805 ***	8.216 ***
Observations	2195	2195	2195	2195	2195	2195	2195	2195	2195
R-squared	0.129	0.261	0.436	0.137	0.257	0.434	0.076	0.252	0.439

*** indicate statistical significance at 1%, 5% and 10%, respectively.

**Table 3 ijerph-17-07422-t003:** Impact of disaster frequency on Jinshi rate, OLS estimates.

Variables	R1	R2	R3
Disaster	0.505 ***	-	-
	(0.165)	-	-
Flood	-	0.270	-
	-	(0.318)	-
Drought	-	-	1.324 ***
	-	-	(0.282)
Area	0.080 *	0.016	0.087 **
	(0.046)	(0.050)	(0.042)
Constant	−9.441 ***	−8.751 ***	−9.589 ***
	(0.404)	(0.436)	(0.351)
Observations	1321	1321	1321
R-squared	0.005	0.001	0.012

***, ** and * indicate statistical significance at 1%, 5% and 10%, respectively. Government financial foundation in that area; ln(pop) and ln(area) illustrate population and area in that county, respectively.

**Table 4 ijerph-17-07422-t004:** Impact of 20 years (1980–2000) of disaster frequency on contemporary human capital, OLS estimates.

Variable	R1	R2	R3	R4	R5	R6
Disaster	1.386 ***	0.395 **	-	-	-	-
	(0.130)	(0.167)		-	-	-
Flood	-	-	0.475 ***	−0.043	-	-
	-	-	(0.149)	(0.195)	-	-
Drought	-	-	-	-	0.760 ***	0.453 ***
	-	-	-	-	(0.118)	(0.172)
Per_gdp	0.064 ***	0.038	0.070 ***	0.037	0.051 **	0.036
	(0.022)	(0.025)	(0.022)	(0.025)	(0.024)	(0.025)
Gender	−0.030 ***	−0.021 ***	−0.032 ***	−0.021 ***	−0.029 ***	−0.021 ***
	(0.005)	(0.005)	(0.005)	(0.004)	(0.005)	(0.004)
Ethic_rate	−0.021 ***	−0.019 ***	−0.021 ***	−0.019 ***	−0.021 ***	−0.019 ***
	(0.001)	(0.001)	(0.001)	(0.001)	(0.001)	(0.001)
Lnarea	0.203 ***	0.172 ***	0.203 ***	0.176 ***	0.265 ***	0.179 ***
	(0.029)	(0.032)	(0.031)	(0.032)	(0.032)	(0.032)
Per_capin	−0.515 **	−0.255	−0.560 **	−0.251	−0.350	−0.241
	(0.261)	(0.297)	(0.265)	(0.297)	(0.285)	(0.297)
FE—Province	NO	YES	NO	YES	NO	YES
Constant	6.680 ***	8.856 ***	8.183 ***	9.218 ***	7.023 ***	8.853 ***
Observations	2131	2131	2131	2131	2131	2131
R-squared	0.258	0.429	0.223	0.427	0.235	0.429

*** and ** indicate statistical significance at the 1%, 5% and 10%, respectively.

**Table 5 ijerph-17-07422-t005:** Impact of 500 years (1950–2000) of disaster frequency on contemporary human capital, OLS estimates (Winsor).

Variable	R1	R2	R3
Disaster	0.635 **	-	-
	(0.247)	-	-
Flood	-	−0.199	-
	-	(0.438)	-
Drought	-	-	1.254 ***
	-	-	(0.413)
Per_gdp	0.148 ***	0.0351	0.144 ***
	(0.0292)	(0.0261)	(0.0279)
Gender	−0.0292 ***	−0.0284 ***	−0.0285 ***
	(0.00402)	(0.00407)	(0.00401)
Ethic_rate	−0.0148 ***	−0.0154 ***	−0.0147 ***
	(0.00103)	(0.00105)	(0.00101)
Lnarea	0.208 ***	0.196 ***	0.205 ***
	(0.0303)	(0.0306)	(0.0299)
Per_capin	−0.0417	−0.184	0.00380
	(0.298)	(0.310)	(0.278)
FE—Province	YES	YES	YES
Constant	8.993 ***	9.559 ***	8.891 ***
Observations	2118	2113	2116
R-squared	0.417	0.412	0.413

*** and ** indicate statistical significance at 1%, 5% and 10%, respectively.

**Table 6 ijerph-17-07422-t006:** Statistical description of China Labor Dynamics Survey Data (CLDS) 2012.

Variable	Definition	Sample	Mean	Sd.	Min	Max
Hc	Education	4514	8.734	3.108	6	22
Disaster	Disaster frequency	4514	0.597	0.194	0.154	0.898
Flood	Flood frequency	4514	0.314	0.114	0.056	0.501
Drought	Drought frequency	4514	0.283	0.092	0.064	0.495
Age	Age	4514	39.016	14.090	15	79
Gender	Gender	4514	1.444	0.497	1	2
Father_edu	Father’s education	4514	1.177	1.084	0	7
Mother_eduu	Mother’s education	4514	0.631	0.795	0	7
Edu_num	School-children number	4514	0.560	0.780	0	4

**Table 7 ijerph-17-07422-t007:** Impact of disaster frequency on contemporary human education level, OLS estimates.

Variable	R1	R2	R3	R4	R5	R6	R7	R8	R9	R10	R11	R12
	OLS	OLS	OLS	HLM	OLS	OLS	OLS	HLM	OLS	OLS	OLS	HLM
Disaster	1.629 ***	1.612 ***	1.594 ***	1.522 ***	-	-	-	-	-	-	-	-
	(0.208)	(0.202)	(0.201)	(0.323)	-	-	-	-	-	-	-	-
Flood	-	-	-	-	2.232 ***	2.303 ***	2.211 ***	2.155 ***	-	-	-	-
	-	-	-	-	(0.351)	(0.342)	(0.343)	(0.559)	-	-	-	-
Drought	-	-	-	-	-	-	-	-	3.820 ***	3.628 ***	3.710 ***	3.456 ***
	-	-	-	-	-	-	-	-	(0.456)	(0.438)	(0.436)	(0.684)
Age	-	−0.031 ***	−0.029 ***	−0.030 ***	-	−0.031 ***	−0.029 ***	−0.030 ***	-	−0.030 ***	−0.029 ***	−0.030 ***
	-	(0.004)	(0.004)	(0.005)	-	(0.004)	(0.004)	(0.005)	-	(0.004)	(0.004)	(0.005)
Gender	-	−0.511 ***	−0.490 ***	−0.561 ***	-	−0.507 ***	−0.487 ***	−0.561 ***	-	−0.516 ***	−0.492 ***	−0.561 ***
	-	(0.088)	(0.088)	(0.090)	-	(0.089)	(0.089)	(0.090)	-	(0.088)	(0.088)	(0.090)
Father_edu	-	0.449 ***	0.446 ***	0.422 ***	-	0.454 ***	0.452 ***	0.422 ***	-	0.449 ***	0.445 ***	0.422 ***
	-	(0.058)	(0.058)	(0.066)	-	(0.058)	(0.058)	(0.066)	-	(0.058)	(0.058)	(0.066)
Mother_edu	-	0.402 ***	0.433 ***	0.407 ***	-	0.404 ***	0.432 ***	0.406 ***	-	0.391 ***	0.427 ***	0.406 ***
	-	(0.076)	(0.077)	(0.085)	-	(0.077)	(0.077)	(0.085)	-	(0.077)	(0.077)	(0.085)
Edu_num	-	−0.482 ***	−0.490 ***	−0.456 ***	-	−0.479 ***	−0.488 ***	−0.455 ***	-	−0.497 ***	−0.506 ***	−0.462 ***
	-	(0.056)	(0.056)	(0.068)	-	(0.056)	(0.0565)	(0.0678)	-	(0.0560)	(0.0563)	(0.0680)
North/South	NO	NO	YES	YES	NO	NO	YES	YES	NO	NO	YES	YES
Constant	7.761 ***	9.190 ***	9.245 ***	9.409 ***	8.032 ***	9.412 ***	9.480 ***	9.627 ***	7.653 ***	9.137 ***	9.166 ***	9.362 ***
Observations	4514	4514	4514	4514	4514	4514	4514	4514	4514	4514	4514	4514
R-squared	0.010	0.114	0.116	-	0.007	0.111	0.113	-	0.013	0.115	0.118	-

*** indicate statistical significance at 1%, 5% and 10%, respectively.

**Table 8 ijerph-17-07422-t008:** Impact of disaster frequency of subsamples human capital divided by north/south and male/female, OLS estimates.

Variable	R1	R2	R3	R4
Disaster	1.737 ***	1.189 ***	1.646 ***	1.555 ***
	(0.278)	(0.367)	(0.312)	(0.327)
Age	−0.014 ***	−0.038 ***	−0.020 ***	−0.045 ***
	(0.005)	(0.005)	(0.005)	(0.006)
Gender	−0.513 ***	−0.492 ***	-	-
	(0.124)	(0.125)	-	-
North/South	-	-	−0.239 *	−0.355 ***
	-	-	(0.124)	(0.128)
Father_edu	0.378 ***	0.493 ***	0.517 ***	0.365 ***
	(0.065)	(0.062)	(0.063)	(0.065)
Mother_edu	0.401 ***	0.494 ***	0.325 ***	0.537 ***
	(0.088)	(0.095)	(0.092)	(0.092)
Edu_num	−0.295 ***	−0.608 ***	−0.486 ***	−0.507 ***
	(0.085)	(0.077)	(0.080)	(0.081)
Constant	8.281 ***	9.811 ***	8.290 ***	8.901 ***
Observations	1940	2574	2511	2003
R-squared	0.089	0.138	0.098	0.146

*** and * indicate statistical significance at 1%, 5% and 10%, respectively.

## Data Availability

Frequency of long-term natural disasters in China data used to support the findings of this study were deposited in the Harvard Dataverse repository (https://doi.org/10.7910/DVN/BVYYEE). CLDS data are openly available from the China Labor-Force Dynamic Survey (http://css.sysu.edu.cn).

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
