# Peer review of "Do Long-Run Disasters Promote Human Capital in China? —The Impact of 500 Years of Natural Disasters on County-Level Human-Capital Accumulation"

_ijerph, 2020, doi:10.3390/ijerph17207422_

Round 1

Reviewer 1 Report

Brief Summary

The paper presents a statistical analysis of 500 years long data record in China to prove causality between natural disasters and human capital accumulation. In regions with higher natural disaster frequency the expected return of physical capital is longer thus making the investment in human capital more attractive. The higher likelihood of human capital investment in these areas has been embedded in the social culture and can even be found in the form of popular proverbs like ‘a book holds a house of gold’. The manuscript presents multivariable linear regressions accounting for different factors that can influence human capital, e.g. gender ratio, area, population, minority rate, the regression analysis is applied to two different databases and time scales. Results show that the linear regression study both in the period 1500-2000 and 1980-2000 presents a good correlation between frequency of natural disasters and human capital even when ruling out the effect of province, or other correlated factors. Results also show that the effect of droughts is greater than the one of floods.

Broad Comments

The reviewer thanks the authors for the well driven research.

There is a series of syntax mistakes that make the reading and understanding more difficult, more details in the specific comments.

How long lasting you expect the effect of high rate natural disasters be in the human capital accumulation? For instance, in a region with historically frequent floods, the construction of a dam reduces the frequency and magnitude of floods, how many years will it take for the public to forget about the importance of human capital investment? Can you comment on that?

It could be useful for the interested reader to introduce the following reference in the introduction: How Natural Disasters Affect the Evolution of Grain Markets: Evidence from 18th Century China  by Y Li, J Ruan, C Ye, 2018, because of its similarities with the present study.

The introduction of a figure showing a map of China with the incidence of different natural disasters or other relevant geographical data could improve the presentation of the paper.

Specific comments (line by line)

The article data number of pages field is 23 while actual manuscript has 12 pages, (it may not be authors fault).

Line 10. The first sentence of the abstract is a strange manner to start an abstract; it looks like it was cut from a longer text. Can you rephrase it?

Line 14. ‘The two’ which two? The two databases are not yet introduced.

Line 14. 1500-2000 years -> the years 1500-2000

Line 15. Increases -> to increase

Line 41. Is FDI defined before?

Line 67. Is it ross-country or cross-county here?

Lines 79-81. Not clear, reformulate sentence.

Line 141. Restore -> restored

Line 160 maybe -> may

Line 185. Yaun -> yuan

Lines 193-195. Syntax, difficult to understand.

Lines 201-202. Syntax, not clear.

Line 217. OLS definition?

Line 217. Where are R1-R9 defined?

Line 265. Where are R1-R3 defined?

Line 279. ‘so in order to further explain the validity of the theory’: this sentence looks out of place here

Line 328. HLM definition?

Author Response

Dear Editors and Reviewers:
Thank you for your letter and for the reviewers’ comments concerning our manuscript entitled “Do long-run Disasters Promote Human Capital in China? --The impact of 500 years of natural disasters on county-level human capital accumulation”. (ijerph-952445). We thank the reviewers for their positive and constructive feedbacks. Those comments are all valuable and very helpful for revising and improving our paper, as well as the important guiding significance to our researches. We have studied comments carefully and have made correction which we hope meet with approval. Revised portion which use the “Track Changes” function in Microsoft Word and are marked in red in the paper. The main corrections in the paper and the responds to the reviewer’s comments are as flowing:

Responds to the reviewer’s comments:

Response to comment: The manuscript requires substantial English editing and restructuring to make it accessible to a wide audience.

Response: Thanks for MDPI English Editing service, our manuscript was edited by their professional editing. Its certification was attached in updated manuscript. After that, we tried our best to improve the manuscript and made some changes in the manuscript.  We have been carefully checked that the abbreviations have been defined; citations within the text are in the correct format; references at the end of the text are in the correct format.

Reviewer #1: 

  1. Response to comment: How long lasting you expect the effect of high rate natural disasters be in the human capital accumulation?

Response: I am sorry for not being able to highlight the impact of disaster frequency on Jinshi rate of this manuscript in the early version. As a robustness test, this result shows that the effect of high rate natural disasters lasting for a really long time. In addition, with limitation in gathering future data, the specific years which takes for the public to forget cannot be determined.

  1. Response to comment: This paper lacks the reference by Y Li, J Ruan, C Ye, 2018

Response: We thank the reviewer for mentioning the papers of Y Li, J Ruan, C Ye, 2018. This paper is certainly relevant related work, and should be discussed. The paper (with which we were previously familiar but neglected to cite) is important, and gives a helpful point to analyze natural disaster frequency’ impact on market integration. As contribution papers for disaster economy literature, both of us use disaster frequency as independent variable, and explore this factor’s impact on long-term transforming. We hope that our work will contribute to and help spur the further development of long-term culture becoming.

  1. Response to comment: The introduction of a figure showing a map of China with the incidence of different natural disasters or other relevant geographical data could improve the presentation of the paper.

Response: It is really true as reviewer suggested that a figure showing a map of China with the incidence of different natural disasters could improve accessibility to a wide audience. “Figure 1. The Regional differences in the frequency of flood and drought disasters in China during period of 1500-2000. Missing values are shown as blanks in the figure.” in Introduction was added. Figure 1 reports the aggregate regional changes in the frequency of flood and drought in China during period of 1500-2000, which overlaid with county boarders in 2000. If this map is divided into three levels with disaster frequency situations, huge floods and floods are concentrated in the central and southern parts of China. The Gansu, Sichuan, Guizhou, and Yunnan provinces in the west had lower levels of floods and droughts, while the more natural disasters were concentrated in Shanxi, Shandong, and Jiangsu provinces.

  1. Specific comments (line by line)

The article data number of pages field is 23 while actual manuscript has 12 pages (it may not be authors fault).

Response: Due to the font size, the revised manuscript has 14 pages.

Line 10. The first sentence of the abstract is a strange manner to start an abstract; it looks like it was cut from a longer text. Can you rephrase it?

Response: Line 10. “Is there a relationship between the frequency of regional natural disasters and long-term human-capital accumulation?” were revised.

Line 14. ‘The two’ which two? The two databases are not yet introduced.

Response: Line14. “On the basis of empirical data of the fifth census in 2000 and China’s Labor-Force Dynamics Survey in 2012” were revised.

            Line 16. 1500-2000 years -> “the years 1500-2000” was revised.

Line 16. Increases -> “to increase” was revised.

Line 41. Is FDI defined before? -> “foreign direct investment” was revised

Line 200. Is it ross-country or cross-county here? -> “cross-county” was corrected.

Lines 214. Not clear, reformulate sentence.

Response: “In previous literature research, natural calamities were considered to be random and exogenous.” was revised.

Line 668. Restore -> “restored” was revised.

Line 560 maybe -> “may” was revised.

Line 698. Yaun -> “CNY” was revised.

Lines 704-705. Syntax, difficult to understand.

Response: “As a climate disaster is an exogenous shock, especially in ancient days, it directly affects local agricultural outputs, for example, grain yield for example.” was revised.

Lines 710-713. Syntax, not clear.

Response: “This equation adds control variables which capture district-level socio-economic factors including county population, areas, ethnic rates, and county-government fixed assets investments. This formula also includes northern or southern region cohort fixed effects, , defined by where the districts are located. Lastly,  is a random disturbance.” Was revised.

Line 725. OLS definition?

Response: “Ordinary-least-squares (OLS)” was added.

Line 839-843. Where are R1-R9 defined?

Response: The definitions of R1-R9 were added in text Line 839-843.

Line 928-931. Where are R1-R3 defined?

Response: The definitions of R1-R3 were added in text Line 928-931.

Line 1011. ‘so in order to further explain the validity of the theory’: this sentence looks out of place here

Response: It is true that this sentence looks out of place here, so this sentence was deleted with reviewer’s suggestion.

Line 1030. HLM definition?

Response: “the multi-level regression model” was added.

Special thanks to you for your good comments. We tried our best to improve the manuscript and made some changes in the manuscript.  

Reviewer 2 Report

This manuscript has the potential to contribute to knowledge. However, it requires significant revision to make it readable and publishable. The manuscript requires substantial English editing and restructuring to make it accessible to a wide audience.

There is lack of clarity on the motivation for the study. The motivation or justification for the study could be described clearly to improve readers’ understanding. In Figure 1, there are many outliers, you could treat the outliers to improve the relationship, or perhaps, sample some of the data for the correlation.

The manuscript does not appear organized in a more presentable form and that has increased reading difficulty. Materials and methods section lack details and information that could be captured under that section are captured under the results section. What needs to be captured and described under the results section are transferred to the discussion section.

Authors need to reorganize manuscript and ensure its content is properly organized under the appropriate headings. Discussion should focus on the intellectual outcomes and the controlling factors together with the implications of the findings. Scientific contribution should be discussed in the discussion section. In the current manuscript, I see many of materials presented in the discussion should rather be moved to the results section.

Based on these comments and several others, I am unable to recommend publication of the manuscript in its current form. Reconsider major revision and resubmit for publication consideration.

Author Response

Dear Editors and Reviewers:
Thank you for your letter and for the reviewers’ comments concerning our manuscript entitled “Do long-run Disasters Promote Human Capital in China? --The impact of 500 years of natural disasters on county-level human capital accumulation”. (ijerph-952445). We thank the reviewers for their positive and constructive feedbacks. Those comments are all valuable and very helpful for revising and improving our paper, as well as the important guiding significance to our researches. We have studied comments carefully and have made correction which we hope meet with approval. Revised portion which use the “Track Changes” function in Microsoft Word and are marked in red in the paper. The main corrections in the paper and the responds to the reviewer’s comments are as flowing:

Responds to the reviewer’s comments:

Response to comment: The manuscript requires substantial English editing and restructuring to make it accessible to a wide audience.

Response: Thanks for MDPI English Editing service, our manuscript was edited by their professional editing. Its certification was attached in updated manuscript. After that, we tried our best to improve the manuscript and made some changes in the manuscript.  We have been carefully checked that the abbreviations have been defined; citations within the text are in the correct format; references at the end of the text are in the correct format.

Reviewer #2: 

  1. Response to comment: There is lack of clarity on the motivation for the study.

Response: We are very sorry for our incorrect writing with lacking of clarity on motivation. As Introduction started with disaster frequency’s long-term impacts, we have re-written this part according to the Reviewer’s suggestion. Line 210-213 was added as reviewer’s suggestion to clarity on the motivation. “As a result of long-term accumulation, regional human capital is not only affected by short-term events, but also likely to be formed through long-term cultural influences, such as an emphasis on education or risk. The current literature on the long-term social impact of natural disasters has a relatively simple and limited conclusion. Thus, this article explores the long-term impression factors of human capital.”

  1. Response to comment: There are many outliers, you could treat the outliers to improve the relationship, or perhaps, sample some of the data for the correlation.

Response: Considering the Reviewer’s suggestion, we have added a robust test which winsorizes the outliers (Table 5).  Impact of 500 years (1950-2000) disaster frequency on contemporary human capital, OLS estimates (Winsor). When winsorization is applied to outliners of the main samples at 1st to 99th percentile (Table 5), R1 indicates the result of fixed effect model, which is efficiently and properly predicting the relationship among disasters and human-capital as R2 value is recorded to be 0.417. The result in R3 shows that winsorize drought has a positive and significant effect on human-capital at 1% confidence level (Line 870-874).

  1. Response to comment: Authors need to reorganize manuscript and ensure its content is properly organized under the appropriate headings.

Response: We are very sorry for our negligence of irregular template in early version. Thanks for peer editors’ kindly arranged reviewer version. While, there were some mistakes of “2. Materials and Methods”, “3. Results” and “4. Conclusion” in reviewer version and the revised version in the template is confirmed with headings and subheadings. We apologize for the decrease in readability due to the incorrect correspondence between headings and contents. In this revised version of manuscript, materials were presented in the correct sections. There is another thing that needs to be reminded. This manuscript combines the discussion part with the conclusion part. Therefore, what should be in the discussion part (e.g. intellectual outcomes and scientific contributions) is actually shown in the conclusion part.

We appreciate for Editors/Reviewers’ warm work earnestly, and hope that the correction will meet with approval.

Once again, thank you very much for your comments and suggestions.

Round 2

Reviewer 2 Report

You have made several changes that have improved the manuscript. I have a couple of concerns. You appear to have multiple objectives in the introduction section of the manuscript. These multiple objectives appear disjointed from each other and there is lack of clarity. It will be great to have them connect to each other and to make the objective(s) more readable.

You mentioned that you performed an expert editing of the manuscript and made some changes afterwards. Several sections of the manuscript require editing to reduce reading difficulty. I will suggest you give the manuscript out for text editing after your changes have been made.

Author Response

Dear Reviewer,

On behalf of my co-authors, we thank you again for giving us second opportunity to revise our manuscript entitled “Do long-run Disasters Promote Human Capital in China? --The impact of 500 years of natural disasters on county-level human capital accumulation” (ijerph-952445). We appreciate your opinions on Introduction and reminders of syntax mistakes in original versions.  We have checked syntax carefully in original version within 2 days and have made correction which we hope meet with approval. Revised portion which use the “Track Changes” function in Microsoft Word and are marked in the paper.

  1. Response to comment: Several sections of the manuscript require editing to reduce reading difficulty.

Response: In this inspection, we found that there were still some inappropriate “the” and other grammatical errors in the original version. We apologize for the repeated occurrence of such errors. The current version has corrected these syntax mistakes. Thank you again for reminding!

  1. Response to comment: You appear to have multiple objectives in the introduction section of the manuscript. These multiple objectives appear disjointed from each other and there is lack of clarity.

Response: It is true that the introduction part in last version has logical confusion. Thank you again for giving us another chance to adjust this part. In this version, the introduction part has 6 paragraphs. Beginning with the important impacts of natural disasters in worldwide, the first paragraph abstracted its impact in ancient China and existing research and shortcomings. After that, two distinct streams of literatures in political economy showed in second paragraph, which examined the impacts of long-run natural disasters on human capital. Then, third paragraph reflected this articles’ innovations, including empirical evidences in agricultural channels and 500 years’ disaster frequency dataset shown in the China map. Additionally, fourth section provided theoretical explanations in the perspectives of psychology and local culture. Moreover, findings and an initial evidence presented by a figure in fifth part. Finally, the last paragraph organized the rest parts of this paper. We have done our best to complete the logical of the introduction part, hoping to meet your requirements. I express my respect for your rigorous academic and strict requirements.

Special thanks to you for your sincere comments. We tried our best to improve the manuscript and made some changes in the manuscript.  

Once again, thank you very much for your comments and suggestions.